# Does Economic Growth Lead to an Increase in Cultivated Land Pressure? Evidence from China

**Xi Wu [1], Yajuan Wang [2] and Hongbo Zhu [2],***

1 School of Economics, Sichuan University, Chengdu 610065, China
2 Department of Land Resource and Real Estate Management, School of Public Administration, Sichuan University, Chengdu 610065, China
* Correspondence: zhb@scu.edu.cn

**Abstract:** With economic growth, people's living standards improve, and more cultivated land is needed to meet food demand. Meanwhile, the economic growth and urban expansion in China since 1978 has led to the loss of considerable amounts of cultivated land. Thus, the contradiction between "economic growth" and "food security" becomes increasingly prominent. Studying the impact of economic growth on cultivated land population support pressure is the basis for easing this problem. This study uses the cultivated land pressure index to represent cultivated land population support pressure, and explores the relationship between economic growth and cultivated land pressure based on the panel data of 31 provinces in China from 2000 to 2017. The feasibility generalized least squares estimation and the fixed effect model based on Driscoll and Kraay standard errors are used. The results show that: (1) the impact of economic growth on cultivated land pressure is an N-shaped or U-shaped curve; and (2) there are regional differences in the impact of economic growth on cultivated land pressure. The cultivated land pressure in economically developed regions and main grain production regions responds slowly to the impact of economic growth. Therefore, some policy recommendations are put forward, such as paying attention to cultivated land protection and controlling disorderly urban expansion.

**Keywords:** economic growth; cultivated land pressure; food security; Kuznets curve





## 1. Introduction

Food is the foundation of human survival and development, and food security attracts worldwide attention [1]. Food production is inseparable from cultivated land, and sufficient cultivated land is an important foundation for ensuring food security [2,3]. However, rapid economic growth and urbanization consume a large amount of cultivated land, which leads to a decrease in cultivated land and a threat to food security [4,5]. As a country with a large population and little cultivated land, China's food security has attracted considerable attention. In 1995, Lester R. Brown published a report entitled "Who Will Feed China?", which alerted people to pay attention to the food security and cultivated land pressure [6]. Since then, scholars have increased their research in related fields [7–9].

At the end of 2017, China's cultivated land area was 134.88 million hm$^2$, ranking third in the world [10]. However, China is the country with the largest population in the world. According to the statistics of FAO, China successfully feeds 19.25% of the global population with only 8.61% of the global cultivated land. In 2017, the global per capita cultivated land area was 0.18 hm$^2$, while this index was only 0.096 hm$^2$ in China [3]. China's cultivated land is under great pressure to support its population. In addition, over the past 40 years, China experienced rapid urbanization and economic growth, which exacerbated food insecurity in China. From 1978 to 2017, the GDP increased from 367.87 billion yuan (USD 21.85 billion at the exchange rate of 1978) to 83,203.59 billion yuan (USD 12,323.17 billion at the exchange rate of 2017), and the urbanization rate increased from 17.92% to 60.24% in China. A large number of studies show that urban expansion would encroach on cultivated land [11,12].

This phenomenon is more pronounced in developing countries, such as China, Vietnam, and India [13–15]. Statistics from the Ministry of Housing and Urban–Rural Development in the People's Republic of China show that 13,258.14 km$^2$ of cultivated land was occupied by urban construction in China from 2000 to 2017.

In recent years, global food insecurity increases significantly under the influence of the COVID-19 pandemic, the Russia–Ukraine conflict, weather extremes, and water scarcity [16–19]. The latest edition of the "State of Food Security and Nutrition in the World" report notes that almost 924 million people faced severe levels of food insecurity in 2021, 207 million more than in 2019 [20]. Under the unstable international situation, trade is restricted, and nations relying on imports are vulnerable to food supply shocks [17]. The statistics of FAO show that China is one of the top ten cereal importers in the world, and its cereal imports in 2020 were about 20% lower than in 2019. The Chinese government begins to advocate using its own cultivated land to feed its population. Xi Jinping, the president of the People's Republic of China, says that "The rice bowls of Chinese people must always be held in their own hands, and the rice bowls are mainly filled with Chinese grains". It is particularly important to coordinate the relationship between economic growth and cultivated land pressure in China. However, the grain supply capacity in different regions of China is diverse. Regions with economic development and high grain production have stronger grain supply capacity and greater grain supply flexibility. The pressure of cultivated land population support may be less affected by economic growth.

Most studies on the relationship between economic growth and cultivated land pressure are based on the Kuznets curve. A Kuznets curve means that the relationship between two variables is an "inverted U", which refers to the way that as one variable increases, the other variable shows a trend of rising first and then falling. In 1955, Simon Kuznets put forward the hypothesis that the relationship between economic growth and wealth distribution takes an inverted U-shaped curve at the Annual Conference of American Economics [21]. In 1991, Grossman introduced the Kuznets curve into the study of the relationship between economic growth and environmental pollution, and put forward the environmental Kuznets curve (EKC) [22]. Since then, scholars have carried out considerable verification and generalization of the traditional inverted U-shaped EKC, and have proposed various shapes of EKC, such as U-shaped, N-shaped, and inverted-N-shaped [23–29]. The research applications are extended to deforestation, ecological footprint, land use, and other aspects [30–36]. Cultivated land has both production and ecological functions, and it is a valuable natural resource. Converting too much cultivated land into construction land would damage the ecological environment. Some scholars believe that the impact of economic growth on cultivated land pressure first rises and then falls, which is similar to the environmental Kuznets curve (EKC). Qu is the first to propose the hypothesis that there is an "inverted U" Kuznets curve between economic growth and farmland conversion [37]. Many studies verify the "inverted U" and "inverted N" Kuznets curves between economic growth and cultivated land conversion based on the provincial panel data in China [37–40]. However, some scholars believe that the existence of a cultivated land Kuznets curve is limited by time and space, and it is not universal [41]. There are monotonically increasing, monotonically decreasing, U-shaped, N-shaped, and inverted N-shaped curves between economic growth and cultivated land conversion [42].

Existing studies only focus on the impact of economic growth on cultivated land loss [38,40,43], without further considering the food security risks and population support pressure caused by cultivated land loss. Based on this, the cultivated land pressure index is used to represent the pressure of cultivated land population support [44]. Then, the impact of economic growth on cultivated land pressure can be studied. It not only enriches the existing research in theory, but also provides new ideas for formulating cultivated land protection strategies and alleviating cultivated land pressure.

Based on EKC hypothesis and the cultivated land pressure index model, this paper studies the impact of economic growth on cultivated land pressure. The main concerns are as follows: (1) whether economic growth increases cultivated land pressure; and

(2) whether there are regional differences in the impact of economic growth on cultivated land pressure. Compared with the existing research, this paper has two innovations. Firstly, the influence path of economic growth on cultivated land pressure is analyzed theoretically. Secondly, the cultivated land pressure index is used to reflect the pressure of cultivated land food security and population support in the empirical study. This research provides a theoretical basis and practical direction for realizing the "double guarantee" of economic growth and food security.

## 2. Materials and Methods

### 2.1. Theoretical Analysis

Research shows that the possible causes of an environmental Kuznets curve (EKC) include the equity of income distribution, international trade, structural changes, technological progress, government governance, and consumer preferences [45]. Cultivated land is an important resource in the environment. Economic structural changes could alter the area of cultivated land occupied by construction. Technological progress could improve land use efficiency. Government policy improvement could restrain the loss of cultivated land, and changes in residents' preferences could increase attention on the ecological function of cultivated land. Some studies have confirmed the influence of these factors [37,46]. Therefore, this paper analyzed the influence of economic growth on cultivated land pressure from the above four aspects.

(1) Economic structural changes. In the era of the agricultural economy, cultivated land was an important means of production. Cultivated land was effectively protected, and cultivated land pressure was small [47]. In the early stage of the industrial economy, land became a key factor to promote economic growth [48]. Urbanization and industrialization transformed large amounts of high-quality cultivated land into construction land [49]. Cultivated land pressure increased rapidly [50]. In the later stage of the industrial economy, land was gradually replaced by capital and labor [51]. The demand for construction land decreased, and cultivated land pressure began to decrease. China entered the later stage of industrialization in 2010 [52], and the area of land requisitioned for construction decreased after reaching the maximum value of 2161.48 km$^2$ in 2012.

(2) Technological progress. In the early stage of economic development, the technological level was low. The proportion of land elements in industrial production was high, and the construction occupied a large amount of cultivated land. Moreover, the level of agricultural technology was also low, and the grain yield per unit area was low. Thus, cultivated land pressure was great. With the advancement of technology, the input of land elements required for economic growth decreases [53], and the grain yield per unit area and land reclamation technology improves [54]. Cultivated land pressure gradually eases. From 2004 to 2017, China's industrial land use efficiency increased from 0.457 to 0.599 [55]. During the same period, the grain yield per unit area of cultivated land in China increased from 4266.94 kg/hm$^2$ to 5607.36 kg/hm$^2$.

(3) Government policy improvement. The focuses of government policies are diverse in different stages of economic and social development. In the beginning stage of reform and opening up, the Chinese government paid attention to economic growth rather than cultivated land protection. With the increasingly serious environmental problems brought by development, the government pays more attention to the ecological environment and sustainable development [56]. In 1998, the "Regulations on the Protection of Basic Farmland" and "Balance between the Occupation and Supplement of Arable Land" were issued, and cultivated land protection measures were gradually tightened [57]. Since then, the Chinese government has issued many policies to continuously strengthen the protection of cultivated land, which effectively control the population support pressure caused by the reduction of cultivated land [4].

(4) Changes in public environmental preferences. The EKC and Inglehart's subjective values hypothesis suggest that as the economy grows, people's priorities shift from economics and materialism to quality of life and subjective wellbeing [58]. Cultivated

land has various ecological functions, such as improving the environment and protecting biodiversity [59,60]. With economic development and income growth, the cultivated land protection gradually attracts public attention.

Based on the analysis, it can be found that the impacts of factors, such as economic structure changes, technological progress, government policy improvement, and public preference changes, on cultivated land pressure are sometimes positive and sometimes negative. Therefore, the relationship between economic growth and cultivated land pressure might be similar to the Kuznets curve. In addition, the territory of China is very vast. There are great differences in the economic development levels and cultivated land-reserve resources in distinct regions. The economically developed regions are mainly distributed on the eastern coast. These regions have limited grain output and are the main grain sales regions. The economically underdeveloped regions are mainly distributed in the central and western regions. The central regions have a flat terrain and are the main grain producing region. The land in the western regions is poor, and most provinces are grain production and sales balance regions. The impact of economic growth on cultivated land pressure may be different in the regions with distinct levels of economic development and grain production and sales.

## 2.2. Regional Division

Based on theoretical analysis, there are differences in the influence of economic growth on cultivated land pressure in the regions with different economic development levels. In addition, China has a vast territory, and the grain production capacity of different provinces is diverse. The cultivated land pressure in main grain production regions might be less affected by economic growth. Therefore, when analyzing the regional differences in the impact of economic growth on cultivated land pressure, the 31 provinces were divided according to the degree of economic development and the situation of grain production and sales. Referring to Tang (2021) [61], the provinces were divided into developed regions and undeveloped regions based on the median of the average per capita GDP from 2000 to 2017. According to the "National Food Security and Long-Term Planning Framework (2008-2020)" proposed by the China National Development and Reform Commission, the provinces were divided into three categories, including the main grain sales regions, the grain production and sales balance regions, and the main grain production regions. The spatial distributions of each region are shown in Figure 1.

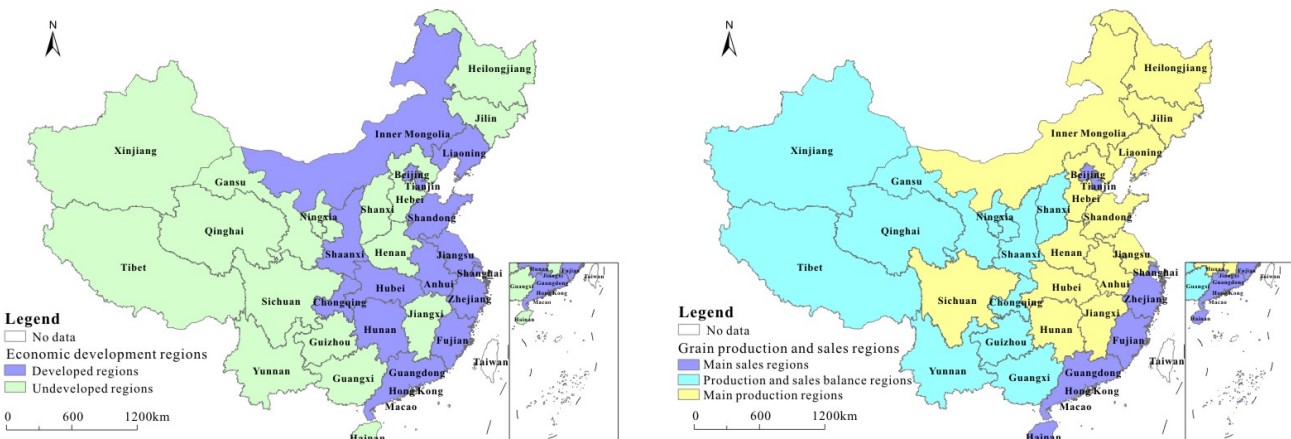

**Figure 1.** The spatial distributions of regions with different levels of economic development and grain production and sales.

### 2.3. Models and Variables

2.3.1. Model Settings

Theoretical analysis shows that the impact of economic growth on cultivated land pressure might be positive first and then negative. This is in line with the characteristics of the environmental Kuznets curve (EKC) model, in that the influence direction of the independent variable changes after reaching a certain value. Most studies applying Kuznets curve model employ reduced-form models, in which the explained variable is the quadratic or cubic function of the explanatory variable [62–65]. Simplified EKC models can clearly specify the form of variable relationships and provide empirical explanations for the solution of research problems [66]. However, the model also has limitations. Firstly, the model only reflects the correlation rather than the causality, and there may be a reverse causality problem in the actual situation [67]. Secondly, the symmetry of quadratic function makes the slope of the uphill and downhill parts of the curve the same, which hardly exists in reality. In addition, the shape of the curve and the number of turning points are affected by the model form. Therefore, the quadratic and cubic function models were established to reduce the fitting error caused by the function form. Since the data of 31 specific provinces in China were used, the following fixed effect model was established:

$$CLP_{it} = \alpha + \beta_1 PGDP_{it} + \beta_2 PGDP_{it}^2 + \beta_i X_{it} + \delta_i + \lambda_t + \mu_{it} \tag{1}$$

$$CLP_{it} = \alpha + \beta_1 PGDP_{it} + \beta_2 PGDP_{it}^2 + \beta_3 PGDP_{it}^3 + \beta_i X_{it} + \delta_i + \lambda_t + \mu_{it} \tag{2}$$

where $i$ and $t$ represent the provinces and periods under consideration; $CLP_{it}$ is the cultivated land pressure index; $PGDP_{it}$ is the per capita GDP; $\alpha$ is a constant; $\beta_1$, $\beta_2$, $\beta_3$, $\beta_i$ are the coefficients to be estimated; $X_{it}$ are control variables, including population ($POP_{it}$), urbanization rate ($UR_{it}$), proportion of secondary industry ($SI_{it}$), proportion of tertiary industry ($TI_{it}$), effective irrigation rate ($EI_{it}$), fertilizer application ($FA_{it}$), pesticide input ($PI_{it}$), and agricultural machinery power ($MP_{it}$). $POP_{it}$ and $UR_{it}$, respectively, reflect the impacts of population growth and urban expansion on cultivated land pressure. $SI_{it}$ and $TI_{it}$ reflect the impacts of industrial structure change on cultivated land pressure. $EI_{it}$, $FA_{it}$, $PI_{it}$ and $MP_{it}$ reflect the impacts of agricultural cultivation level and technology on cultivated land pressure. $\delta_i$ and $\lambda_t$, respectively, denote the region and time effects. $\mu_{it}$ is a random error term.

According to different situations of estimation coefficients $\beta_1 - \beta_3$, the different shapes and possible turning points of the Kuznets curve are shown in Table 1.

**Table 1.** Possible results of the cultivated land pressure Kuznets curve model.

| Function Type | $\beta_1$ | $\beta_2$ | $\beta_3$ | Curve Shape | Possible Turning Points |
|---|---|---|---|---|---|
| Cubic function | — | — | >0 | N or monotonically increasing | $\frac{-\beta_2 \pm \sqrt{\beta_2^2 - 3\beta_1\beta_3}}{3\beta_3}$ |
| | — | — | <0 | Inverted N or monotonically decreasing | |
| Quadratic function | — | >0 | =0 | U | $-\frac{\beta_1}{2\beta_2}$ |
| | — | <0 | =0 | Inverted U | |
| Linear function | >0 | =0 | =0 | Monotonically increasing | — |
| | <0 | =0 | =0 | Monotonically decreasing | — |

2.3.2. Variables Selection

(1) Explained variable. Cultivated land pressure index was used to characterize the cultivated land pressure, which is proposed by Cai Yunlong (2002) [44]. It takes into account the food demand of the population, the grain production capacity, and the area of cultivated land, and it can comprehensively reflect the pressure of cultivated land to ensure population support in a certain region. The cultivated land pressure index is the ratio of

the minimum per capita cultivated land area to the actual per capita cultivated land area, and the basic calculation formula of it is as follows:

$$K_i = \frac{S_{\min i}}{S_i} = \frac{\beta_i \times \frac{Gr_i}{p_i \cdot q_i \cdot k_i}}{S_i} \qquad (3)$$

where $K_i$ is the cultivated land pressure index. $S_{\min i}$ is the minimum per capita cultivated land area, which refers to the area of cultivated land required to ensure the normal food consumption of each person under a certain level of grain self-sufficiency and cultivated land production capacity in a certain region ($S_{\min i} = \beta_i \times Gr_i / (p_i \cdot q_i \cdot k_i)$). $S_i$ is the actual per capita cultivated land area, which is the ratio of the total cultivated land area to the total population in a region. $\beta_i$ is the grain self-sufficiency rate, which refers to the proportion of grain production to grain consumption in the region. $Gr_i$ is the per capita grain demand, usually calculated based on calories consumed or statistics [68,69]. $p_i$ is the grain yield per unit area. $q_i$ is the proportion of grain crop sown area in the total crop sown area. $k_i$ is the multiple cropping index, which represents the ratio of crop sown area to cultivated land area within a year. When $K_i < 1$, the cultivated land grain production is greater than the demand, and there is no cultivated land pressure. When $K_i = 1$, the cultivated land grain production is equal to the demand, and cultivated land pressure is at a critical value. When $K_i > 1$, the cultivated land grain production is less than the demand, and there is cultivated land pressure.

Due to the different levels of economic development, the relationship of grain production and sales among provinces is different. That is to say, there are differences in the economic acquisition capacity of grain in distinct provinces. Referring to Zhu (2016) [70], the first revision of the cultivated land pressure index was carried out by using the economic acquisition capacity of grain. In addition, there are differences in the quality of cultivated land in distinct provinces. Referring to Luo (2016) [71], the second revision of the cultivated land pressure index was carried out by using the standard coefficient of cultivated land productivity. The calculation formula of the revised cultivated land pressure index is as follows:

$$K_i{}' = K_i \times \frac{1}{\theta_i} \times \frac{1}{\sigma_i} = \frac{\beta_i \times \frac{Gr_i}{p_i \cdot q_i \cdot k_i}}{S_i} \times \frac{\overline{X}}{X_i} \times \frac{p \cdot k}{p_i \cdot k_i} \qquad (4)$$

where $K_i{}'$ is the revised cultivated land pressure index. $\theta_i$ is the grain economic acquisition capacity of province $i$, which is expressed by the ratio of the per capita GDP of province $i$ to that of the nation ($\theta_i = X_i / \overline{X}$). $\overline{X}$ is the national average per capita GDP. $X_i$ is the per capita GDP of province $i$. $\sigma_i$ is the standard coefficient of cultivated land productivity, which is expressed by the ratio of the cultivated land production capacity of province $i$ to that of the nation ($\sigma_i = (p_i \cdot k_i) / (p \cdot k)$). $p$ is the national grain yield per unit area. $k$ is the national multiple cropping index. The meanings of the other indicators are the same as those in formula (3).

(2) Explanatory variable. The explanatory variable of this paper is economic growth. Existing studies mostly use indicators such as GDP, per capita GDP, and GDP growth rate to characterize economic growth [72–75]. Among them, per capita GDP can better reflect the average level of regional economic growth. In recent years, China's economy has developed rapidly, and both population and GDP has grown. Thus, per capita GDP was used to represent economic growth.

(3) Control variables. In the process of economic growth, other factors can affect the pressure of cultivated land population support. Theoretical analysis shows that industrial structure changes and agricultural technology progress would affect cultivated land pressure. Some studies have confirmed the impact of population growth and urbanization on cultivated land [76–78]. In recent years, China's major industries transforms from the secondary industry to the tertiary industry [79]. Firstly, the development of non-agricultural industries may occupy cultivated land, which results in the reduction of cultivated land. Secondly, it may promote the labor force to leave agricultural production and reduce the ef-

ficiency of grain production [80]. In addition, agricultural production technology is rapidly improved, agricultural irrigation and mechanization are popularized, and the inputs of fertilizer and pesticide are increased. The above factors have a significant impact on ensuring the quantity and productivity of cultivated land [76]. Therefore, when analyzing the factors affecting cultivated land pressure, eight control variables were selected, including population, urbanization rate, proportion of secondary industry, proportion of tertiary industry, irrigation rate, fertilizer application, pesticide input, and agricultural machinery power. The explanation of the variables is shown in Table 2.

**Table 2.** Explanation of the variables.

| Variable Types | Variable Names | Variable Connotation | Unit |
|---|---|---|---|
| Explained variable | Cultivated land pressure (CLP) | Cultivated land pressure index | — |
| Explanatory variable | Economic growth (PGDP) | Per capita GDP (at the price in 2000) | $10^4$ yuan/person |
| | Population (POP) | Total population | $10^8$ persons |
| | Urban expansion (UR) | Urban population/total population | % |
| | Proportion of secondary industry (SI) | Added value of secondary industry/GDP | % |
| | Proportion of tertiary industry (TI) | Added value of tertiary industry/GDP | % |
| Control variables | Effective irrigation rate (EI) | Effective irrigation area/cultivated land area | % |
| | Fertilizer application (FA) | Fertilizer application/cultivated land area | $10^4$ t/hm$^2$ |
| | Pesticide input (PI) | Pesticide input/cultivated land area | $10^4$ t/hm$^2$ |
| | Agricultural machinery power (MP) | Agricultural machinery power/cultivated land area | KW/hm$^2$ |

### 2.4. Data Sources

Since China conducted the third national land survey in 2017, the data of cultivated land area has not been continuously updated. Therefore, the panel data of 31 provinces (excluding Hong Kong, Macao, and Taiwan) in China from 2000 to 2017 were used.

The level of economic development is expressed by per capita GDP (PGDP). The consumer price index (CPI) was used to convert the per capita GDP into a comparable price in 2000. The data on the grain yield per unit area, grain crop sown area, total crop sown area, cultivated land area, population, urbanization rate, proportion of secondary industry, proportion of tertiary industry, irrigation rate, fertilizer application, pesticide input, agricultural machinery power, GDP, and CPI were obtained from the "China Statistical Yearbook (2001–2018)" and the "Provincial Statistical Yearbook". Referring to the existing research, the grain self-sufficiency rate was set as 1 [81]; the per capita grain demand was set as 350 kilos per person in 1981, with an increase of 4 kg per year after 1981 and a decrease of 4 kg per year before 1981 [70]. The descriptive statistics for the data are illustrated in Table 3.

**Table 3.** Descriptive statistics of the variables.

| Variable Names | Mean | Std. Dev. | Min. | Max. | Obs. | Skewness | Kurtosis |
|---|---|---|---|---|---|---|---|
| CLP | 2.2487 | 2.1445 | 0.3447 | 21.3217 | 558 | 2.672 | 16.051 |
| PGDP | 2.2412 | 1.6428 | 0.2742 | 9.9292 | 558 | 1.629 | 6.118 |
| POP | 0.4273 | 0.2734 | 0.0258 | 1.2141 | 558 | 0.608 | 2.606 |
| UR | 48.8493 | 15.9550 | 19.4700 | 89.6000 | 558 | 0.579 | 3.057 |
| SI | 42.9756 | 8.2835 | 16.8972 | 61.9603 | 558 | −0.719 | 3.508 |
| TI | 44.5186 | 8.6279 | 29.6445 | 82.6948 | 558 | 1.769 | 7.639 |
| EI | 50.6619 | 22.5083 | 13.6963 | 115.2961 | 558 | 0.411 | 2.118 |
| FA | 0.0431 | 0.0215 | 0.0068 | 0.1001 | 558 | 0.391 | 2.485 |
| PI | 0.0015 | 0.0013 | 0.0001 | 0.0065 | 558 | 1.167 | 4.029 |
| MP | 0.6870 | 0.3773 | 0.1297 | 1.7545 | 558 | 0.725 | 2.547 |

The correlation matrix of the variables and the variance expansion factor (VIF) of the multicollinearity tests are shown in Table 4.

**Table 4.** The correlation matrix of the variables and the results of the multicollinearity tests.

| Variables | CLP | PGDP | POP | UR | SI | TI | EI | FA | PI | MP | VIF |
|---|---|---|---|---|---|---|---|---|---|---|---|
| CLP | 1.000 | — | — | — | — | — | — | — | — | — | — |
| PGDP | 0.042 | 1.000 | — | — | — | — | — | — | — | — | 5.66 |
| POP | −0.419 *** | 0.025 | 1.000 | — | — | — | — | — | — | — | 2.12 |
| UR | −0.113 *** | 0.849 *** | −0.079 * | 1.000 | — | — | — | — | — | — | 4.60 |
| SI | −0.400 *** | −0.091 ** | 0.446 *** | 0.026 | 1.000 | — | — | — | — | — | 4.38 |
| TI | 0.399 *** | 0.629 *** | −0.395 *** | 0.530 *** | −0.699 *** | 1.000 | — | — | — | — | 7.21 |
| EI | −0.325 *** | 0.541 *** | 0.229 *** | 0.446 *** | 0.092 ** | 0.275 *** | 1.000 | — | — | — | 2.29 |
| FA | −0.369 *** | 0.430 *** | 0.543 *** | 0.353 *** | 0.215 *** | 0.018 | 0.612 *** | 1.000 | — | — | 4.27 |
| PI | −0.214 *** | 0.332 *** | 0.320 *** | 0.297 *** | 0.063 | 0.047 | 0.472 *** | 0.764 *** | 1.000 | — | 2.61 |
| MP | −0.132 *** | 0.404 *** | 0.352 *** | 0.294 *** | 0.176 *** | 0.176 *** | 0.620 *** | 0.533 *** | 0.364 *** | 1.000 | 1.99 |
| Mean VIF | — | — | — | — | — | — | — | — | — | — | 3.90 |

Note: *, **, and *** indicate the significance of 10%, 5% and 1%, respectively.

## 3. Empirical Results

### 3.1. Unit Root Tests

The unit root test can prevent spurious regression by testing the stationarity of panel data [82]. Depending on the null hypothesis, unit root tests can be divided into two categories. The first type assumes that each section has the same unit root, including the LLC (Levin–Lin–Chu) test and the Breitung test. The second type assumes that each section has a different unit root, including the IPS (Im–Pesaran–Shin) test, the Fisher-ADF test and the Fisher-PP test. In this paper, four methods are used to test the unit root. The results of the unit root test show that the variables are first-order stable (Table 5), and it is valid to perform regression analysis.

**Table 5.** Results of unit root tests.

| Variables | LLC Test | IPS Test | Fisher−ADF Test | Fisher−PP Test |
|---|---|---|---|---|
| d(CLP) | −21.306 *** | −18.720 *** | 427.301 *** | 851.230 *** |
| d(PGDP) | −5.490 *** | −3.635 *** | 101.489 *** | 83.893 *** |
| d(POP) | −7.136 *** | −6.107 *** | 149.611 *** | 146.907 *** |
| d(UR) | −10.539 *** | −9.226 *** | 208.395 *** | 321.182 *** |
| d(SI) | −8.111 *** | −5.680 *** | 139.542 *** | 202.138 *** |
| d(TI) | −10.461 *** | −7.930 *** | 173.880 *** | 164.494 *** |
| d(EI) | −19.443 *** | −14.645 *** | 300.356 *** | 446.513 *** |
| d(FA) | −10.566 *** | −8.908 *** | 193.441 *** | 224.793 *** |
| d(PI) | −9.748 *** | −9.964 *** | 227.856 *** | 263.869 *** |
| d(MP) | −14.500 *** | −11.045 *** | 229.555 *** | 245.916 *** |

Note: *** indicates the significance of 1%.

### 3.2. Basic Estimation Results

In order to ensure the reliability of the regression results, the Hausman test and F statistic are used for model selection. According to the test results, the fixed-effects model is considered to be superior to the random-effects or mixed model. The heteroscedasticity, cross-sectional dependency. and serial correlation tests are necessary for the panel data [83]. The modified Wald test, Frees test, and Wooldridge test are used to check for the above problems, respectively [84–86]. The test results show that the standard fixed-effects model has heteroscedasticity and correlation problems, which may cause estimation inefficiency [87]. Therefore, the estimation method is changed in the robustness test. The basic estimation results are shown in Table 6.

According to the estimation results of the cubic model, the coefficients of $PGDP^3$ are significantly positive at the level of 1%. This shows that with economic growth, the cultivated land pressure increases firstly, then decreases, and increases again finally. There is an N-shaped cultivated land pressure Kuznets curve. According to the estimation results of the quadratic model, the coefficients of $PGDP^2$ are significantly positive at the level of

1%. This shows that with economic growth, the cultivated land pressure first decreases and then increases. When the per capita GDP is about 40,000 yuan/person, the pressure on cultivated land begins to rebound. From 2000 to 2017, the average per capita GDP in each province increased from 8430 yuan/person to 41,270 yuan/person. Hence, the rebound point of cultivated land pressure is approaching.

**Table 6.** The results of basic estimation.

| Variables | Fe_c | Fe_cc | Fe_q | Fe_qc |
|---|---|---|---|---|
| $PGDP^3$ | 0.037 *** (9.663) | 0.033 *** (6.417) | — | — |
| $PGDP^2$ | −0.337 *** (−5.923) | −0.271 *** (−3.287) | 0.193 *** (11.548) | 0.246 *** (13.637) |
| PGDP | 0.856 *** (3.215) | 0.448 (0.962) | −1.157 *** (−6.427) | −2.172 *** (−9.327) |
| POP | 10.968 *** (6.001) | 10.632 *** (5.727) | 4.102 ** (2.243) | 6.441 *** (3.566) |
| UR | −0.008 (−0.690) | −0.014 (−1.104) | 0.010 (0.797) | −0.011 (−0.876) |
| SI | 0.085 *** (4.428) | 0.097 *** (4.820) | 0.137 *** (6.816) | 0.125 *** (6.165) |
| TI | 0.081 *** (3.788) | 0.067 *** (2.848) | 0.124 *** (5.453) | 0.062 ** (2.534) |
| EI | −3.098 *** (−4.400) | −3.206 *** (−4.376) | −3.787 *** (−4.980) | −4.003 *** (−5.334) |
| FA | −27.000 *** (−2.991) | −23.124 ** (−2.469) | −8.406 (−0.878) | −13.375 (−1.392) |
| PI | 453.960 *** (4.400) | 479.197 *** (4.453) | 433.659 *** (3.872) | 452.687 *** (4.050) |
| MP | −1.038 *** (−3.205) | −1.038 *** (−3.079) | −0.637 * (−1.828) | −0.816 ** (−2.341) |
| Cons | −7.228 *** (−4.163) | −6.451 *** (−3.350) | −8.212 *** (−4.364) | −4.110 ** (−2.091) |
| Time−fixed effect | No | Yes | No | Yes |
| Region−fixed effect | Yes | Yes | Yes | Yes |
| $R^2$ | 0.604 | 0.612 | 0.532 | 0.580 |
| Modified Wald test | 46,481.77 *** | 24,021.74 *** | 89,123.87 *** | 28,739.68 *** |
| Frees test | 5.052 *** (0.144) | 4.723 *** (0.144) | 4.836 *** (0.144) | 4.642 *** (0.144) |
| Wooldridge test | 14.076 *** | 13.942 *** | 14.515 *** | 12.895 *** |
| F test | 71.44 *** | 28.12 *** | 58.75 *** | 25.58 *** |
| F statistic | 43.03 *** | 40.47 *** | 34.93 *** | 36.23 *** |
| Hausman test | 118.26 *** | 115.44 *** | 46.22 *** | 48.40 *** |
| Curve shape | N | N | U | U |
| Maximum extreme point | 1.813 | 1.019 | — | — |
| Minimum extreme point | 4.239 | 4.386 | 3.005 | 4.416 |
| Obs. | 558 | 558 | 558 | 558 |

Note: (1) The data outside the brackets are coefficients, and the data inside the brackets are t values; the critical value of 10% significance is shown in the brackets of the Frees test. (2) Fe_c, Fe_q are the estimation results after controlling the region effect; Fe_cc, Fe_qc are the estimation results after controlling the region effect and the time effect. (3) *, **, and *** indicate the significance of 10%, 5%, and 1%, respectively.

The effects of control variables on cultivated land pressure are basically identical in all models. The impact of population on cultivated land pressure is significantly positive at the level of 5%. This is consistent with the research results of other scholars [88]. It shows that population growth increases the demand for food and the space for construction land, which increases the cultivated land pressure. The impact of urbanization is negative, but not significant. This may be due to the offsetting effect between cultivated land abandonment and the increase in the ratio of grain crops caused by the migration of rural population to cities [76]. On the one hand, urban expansion occupies a large amount of cultivated land, which results in the reduction of cultivated land [11,89]. On the other hand, population urbanization leads to the transfer of labor from agricultural industries to non-agricultural industries, which may force the increase of agricultural operation scale and mechanization, and the proportion of grain crops may increase [90]. The coefficients of the proportion of the secondary industry and the proportion of the tertiary industry are significantly positive at the level of 5%. This shows that the increases of secondary and tertiary industries exacerbate the cultivated land pressure. The coefficients of effective irrigation rate, fertilizer application, and agricultural machinery power are significantly negative. This shows that the improvement of agricultural production level and technology can reduce the cultivated land pressure. However, pesticide input has a positive impact on cultivated land pressure. This may be because China's pesticide input has exceeded the economic optimal level [91].

The increase of pesticide input would lead to many adverse effects and increase the pressure on cultivated land [92].

### 3.3. Robustness Analysis

3.3.1. Replacement of Explanatory Variable

The per capita disposable income can reflect the wealth level of residents, and can be used to measure economic growth [93]. Therefore, the per capita disposable income (PDI) of residents is selected as the alternative variable of per capita GDP (PGDP) for the robustness test. The estimation results are shown in Table 7.

**Table 7.** Estimation results of the replacement explanatory variable.

| Variables | Fe_c | Fe_cc | Fe_q | Fe_qc |
|---|---|---|---|---|
| $PDI^3$ | 0.370 *** (6.529) | 0.283 *** (3.670) | — | — |
| $PDI^2$ | −1.354 *** (−3.740) | −0.659 (−1.219) | 0.945 *** (10.843) | 1.289 *** (12.302) |
| PDI | 1.517 ** (2.198) | −0.640 (−0.467) | −2.177 *** (−5.303) | −5.085 *** (−7.819) |
| POP | 9.943 *** (5.132) | 9.732 *** (4.929) | 3.856 ** (2.184) | 6.533 *** (3.642) |
| UR | −0.007 (−0.563) | −0.012 (−0.911) | 0.008 (0.665) | −0.004 (−0.323) |
| SI | 0.092 *** (4.802) | 0.092 *** (4.499) | 0.119 *** (6.112) | 0.094 *** (4.538) |
| TI | 0.085 *** (3.856) | 0.064 *** (2.588) | 0.098 *** (4.255) | 0.055 ** (2.233) |
| EI | −3.359 *** (−4.591) | −3.801 *** (−4.898) | −3.811 *** (−5.034) | −4.470 *** (−5.853) |
| FA | −28.534 *** (−3.010) | −26.084 *** (−2.673) | −11.393 (−1.203) | −18.525 * (−1.918) |
| PI | 526.875 *** (5.023) | 562.301 *** (5.186) | 505.879 *** (4.642) | 556.790 *** (5.073) |
| MP | −1.235 *** (−3.672) | −1.246 *** (−3.549) | −0.803 ** (−2.341) | −1.038 *** (−2.958) |
| Cons | −7.040 *** (−4.022) | −4.912 ** (−2.379) | −6.265 *** (−3.451) | −2.242 (−1.146) |
| Time−fixed effect | No | Yes | No | Yes |
| Region−fixed effect | Yes | Yes | Yes | Yes |
| $R^2$ | 0.574 | 0.584 | 0.539 | 0.573 |
| Modified Wald test | 36,987.34 *** | 20,376.38 *** | 48,921.38 *** | 22,332.70 *** |
| Frees test | 4.225 *** (0.144) | 4.378 *** (0.144) | 4.362 *** (0.144) | 4.457 *** (0.144) |
| Wooldridge test | 10.060 *** | 9.986 *** | 13.568 *** | 12.517 *** |
| F test | 63.22 *** | 25.05 *** | 60.42 *** | 24.86 *** |
| F statistic | 37.11 *** | 40.91 *** | 36.60 *** | 37.84 *** |
| Hausman test | 45.87 *** | 99.69 *** | 48.75 *** | 79.93 *** |
| Curve shape | N | N | U | U |
| Maximum extreme point | 0.872 | −0.388 | — | — |
| Minimum extreme point | 1.568 | 1.941 | 1.152 | 1.972 |
| Obs. | 558 | 558 | 558 | 558 |

Note: (1) The data outside the brackets are coefficients, and the data inside the brackets are t values; the critical value of 10% significance is shown in the brackets of the Frees test. (2) Fe_c, Fe_q are the estimation results after controlling the region effect; Fe_cc, Fe_qc are the estimation results after controlling the region effect and the time effect. (3) *, **, and *** indicate the significance of 10%, 5%, and 1%, respectively.

After replacing the explanatory variable, the estimation results are consistent with basic estimation. The cubic model shows that as per capita disposable income increases, the cultivated land pressure increases firstly, then decreases, and increases again finally. The estimation results of the squared model show that there is a U-shaped curve relationship between per capita disposable income growth and cultivated land pressure. When the PDI is between 15,000–20,000 yuan/person, the cultivated land pressure starts to rebound. From 2000 to 2017, the average per capita disposable income in each province increased from 4010 yuan/person to 17,620 yuan/person. The rebound points of cultivated land pressure are close to basic estimations. The influence direction and significance of the control variables are basically consistent with basic estimation. This shows that the impact of economic growth on cultivated land pressure is stable.

### 3.3.2. Change of Estimation Methods

With the existence of heteroscedasticity, cross-sectional dependence, and autocorrelation, the feasibility generalized least squares (FGLS) technique and Driscoll and Kraay

standard error are employed [61,94]. Driscoll and Kraay standard errors are produced through weighted heteroskedasticity autocorrelation (HAC), which can effectively address the complications caused by heteroscedasticity, cross-sectional dependence, and autocorrelation [87]. The estimation results after changing the estimation methods are shown in Table 8.

**Table 8.** Estimation results after changing the estimation methods.

| Variables | FGLS_c | FGLS_q | Fe_ccd | Fe_qcd |
|---|---|---|---|---|
| PGDP$^3$ | 0.012 ** (2.491) | — | 0.033 *** (4.473) | — |
| PGDP$^2$ | −0.011 (−0.171) | 0.172 *** (9.834) | −0.271 ** (−2.567) | 0.246 *** (6.360) |
| PGDP | −0.467 (−1.553) | −1.448 *** (−8.478) | 0.448 (0.888) | −2.172 *** (−5.755) |
| POP | 5.594 *** (6.033) | 5.004 *** (4.955) | 10.632 *** (9.856) | 6.441 *** (3.356) |
| UR | −0.008 (−1.150) | 0.004 (0.488) | −0.014 ** (−2.339) | −0.011 *** (−2.755) |
| SI | 0.044 *** (3.672) | 0.059 *** (5.025) | 0.097 *** (6.751) | 0.125 *** (8.330) |
| TI | 0.035 *** (2.628) | 0.045 *** (3.266) | 0.067 * (1.742) | 0.062 (1.516) |
| EI | −1.579 *** (−3.634) | −1.326 *** (−2.670) | −3.206 *** (−3.494) | −4.003 *** (−3.523) |
| FA | −12.364 *** (−3.271) | −16.843 *** (−4.529) | −23.124 ** (−2.646) | −13.375 (−1.259) |
| PI | 342.547 *** (3.959) | 349.341 *** (3.988) | 479.197 *** (2.930) | 452.687 *** (2.850) |
| MP | −0.340 ** (−2.038) | −0.165 (−0.909) | −1.038 ** (−2.418) | −0.816 ** (−2.284) |
| Cons | 1.730 (1.389) | 1.319 (0.960) | −5.740 * (−1.789) | −0.681 (−0.173) |
| Time−fixed effect | Yes | Yes | Yes | Yes |
| Region−fixed effect | Yes | Yes | Yes | Yes |
| R$^2$ | — | — | 0.612 | 0.580 |
| F/Wald test | 4404.25 *** | 5169.83 *** | 795.44 *** | 236.26 *** |
| Curve shape | N | U | N | U |
| Maximum extreme point | −3.309 | — | 1.019 | — |
| Minimum extreme point | 3.92 | 4.209 | 4.386 | 4.416 |
| Obs. | 558 | 558 | 558 | 558 |

Note: (1) The data outside the brackets are coefficients, and the data inside the brackets are t values. (2) FGLS_c and FGLS_q are the estimation results with FGLS; Fe_ccd, Fe_qcd demonstrate Driscoll and Kraay standard errors. (3) *, **, and *** indicate the significance of 10%, 5%, and 1%, respectively.

After changing the estimation method, the influence direction and significance of the explanatory variable and control variables are basically consistent with the basic estimation. The N-shaped or U-shaped curve relationship between economic growth and cultivated land pressure is proved to be stable again.

### 3.4. Endogenous Analysis

There are many factors that affect the pressure of cultivated land. Although the basic estimation has controlled the main influencing factors, there are still some factors that have been missed. In addition, there may also be a reverse causal relationship between the explanatory variable and the explained variable. These may lead to endogeneity problems in the model. The generalized moment estimation (GMM) proposed by Arellano and Bond (1991) can deal with endogeneity problems by introducing a lag of explained variables [95]. In this paper, an improved system generalized moment estimation (sys-GMM) is used for endogenous analysis [96]. The endogenous test results are shown in Table 9.

The results of system generalized moment estimation are basically consistent with basic estimation. Only the influence direction and significance of a few control variables change. In addition, the model passes the serial correlation test (the *p* value of AR(1) is less than 0.1, the *p* value of AR(2) is greater than 0.1) and the validity test of instrumental variables (the *p* value of the Hansen test is greater than 0.1) [97]. Therefore, it can be considered that the estimation results are stable and reliable.

**Table 9.** Estimation results with generalized moments.

| Variables | GMM_ct | GMM_qt | GMM_cr | GMM_qr |
|---|---|---|---|---|
| L.CLP | 0.872 *** (84.645) | 0.873 *** (101.989) | 0.868 *** (17.452) | 0.869 *** (17.639) |
| PGDP$^3$ | 0.017 *** (8.869) | — | 0.018 * (1.768) | — |
| PGDP$^2$ | −0.146 *** (−6.203) | 0.070 *** (9.892) | −0.163 (−1.538) | 0.068 * (1.896) |
| PGDP | 0.362 *** (4.989) | −0.393 *** (−7.737) | 0.417 (1.348) | −0.350 * (−1.753) |
| POP | −0.295 *** (−3.313) | −0.406 *** (−5.436) | −0.295 * (−1.699) | −0.372 ** (−2.286) |
| UR | −0.010 *** (−5.296) | −0.009 *** (−5.769) | −0.011 ** (−2.068) | −0.010 ** (−1.973) |
| SI | 0.013 *** (4.454) | 0.019 *** (4.170) | 0.012 (1.432) | 0.013 (1.366) |
| TI | 0.028 *** (8.977) | 0.031 *** (6.181) | 0.028 ** (2.466) | 0.025 * (1.882) |
| EI | −1.057 *** (−10.045) | −1.326 *** (−11.241) | −0.954 *** (−2.638) | −1.218 ** (−2.371) |
| FA | 0.532 (0.249) | 1.313 (1.311) | 0.037 (0.015) | 0.145 (0.074) |
| PI | 77.834 *** (2.864) | 85.714 *** (4.618) | 76.243 ** (2.144) | 78.439 ** (2.134) |
| MP | 0.238 *** (2.843) | 0.391 *** (9.189) | 0.234 * (1.870) | 0.370 ** (2.181) |
| Cons | −0.941 *** (−3.699) | −0.692 * (−1.706) | −0.874 * (−1.679) | −0.179 (−0.247) |
| AR(1) | −2.38 (0.017) | −2.39 (0.017) | −2.56 (0.011) | −2.54 (0.011) |
| AR(2) | 1.11 (0.269) | 1.05 (0.296) | 1.19 (0.234) | 1.10 (0.272) |
| Hansen test | 23.12 (0.145) | 21.01 (0.226) | 23.12 (0.145) | 21.01 (0.226) |
| Curve shape | N | U | N | U |
| Maximum extreme point | 1.802 | — | 1.827 | — |
| Minimum extreme point | 3.979 | 2.814 | 4.246 | 2.560 |
| Obs. | 527 | 527 | 527 | 527 |

Note: (1) The data outside the brackets are coefficients, and the data inside the brackets are t values. (2) GMM_ct, GMM_qt are the results of two-step estimation; GMM_cr, GMM_qr are the results of robust estimation. (3) The *p* values of AR(1), AR(2), and the Hansen test are in parentheses. (4) *, **, and *** indicate the significance of 10%, 5%, and 1%, respectively.

### 3.5. Heterogeneity Analysis

### 3.5.1. Different Economic Development Regions

The estimation results of different economic development regions are shown in Table 10.

**Table 10.** Estimation results for different economic development regions.

| Variables | Developed Regions | | Undeveloped Regions | |
|---|---|---|---|---|
| | FE_ccd | FE_qcd | FE_ccd | FE_qcd |
| PGDP$^3$ | 0.053 *** (6.647) | — | −0.668 *** (−4.885) | — |
| PGDP$^2$ | −0.642 *** (−4.739) | 0.242 *** (5.896) | 4.864 *** (6.652) | 1.245 *** (6.478) |
| PGDP | 3.006 *** (3.597) | −1.898 *** (−5.305) | −14.291 *** (−8.886) | −8.046 *** (−7.558) |
| POP | 15.051 *** (7.225) | 12.546 *** (7.309) | −5.955 ** (−2.720) | −1.804 (−0.699) |
| UR | −0.004 (−0.806) | −0.007 (−1.263) | 0.032 ** (2.570) | 0.003 (0.194) |
| SI | 0.077 ** (2.617) | 0.294 *** (5.092) | 0.179 *** (11.780) | 0.135 *** (10.230) |
| TI | 0.012 (0.442) | 0.127 ** (3.213) | 0.141 *** (5.348) | 0.088 *** (3.505) |
| EI | −3.716 *** (−4.198) | −6.194 *** (−3.421) | 1.225 (0.875) | 0.038 (0.030) |
| FA | −76.620 *** (−4.503) | −48.076 *** (−3.611) | −10.096 (−0.750) | 2.569 (0.171) |
| PI | 1474.904 *** (5.470) | 1022.763 *** (4.473) | −50.447 (−0.668) | 95.053 (1.140) |
| MP | −2.527 *** (−4.674) | −1.636 *** (−3.575) | 0.735 (1.611) | 0.827 (1.701) |
| Cons | 0 | 0 | 5.455 ** (2.271) | 5.793 * (2.122) |
| Time−fixed effect | Yes | Yes | Yes | Yes |
| Region−fixed effect | Yes | Yes | Yes | Yes |
| R$^2$ | 0.827 | 0.771 | 0.415 | 0.365 |
| F test | 1432.66 *** | 291.56 *** | 87,740.96 *** | 5617.11 *** |
| Curve shape | Increment | U | Decrement | U |
| Maximum extreme point | — | — | — | — |
| Minimum extreme point | — | 3.922 | — | 3.231 |
| Obs. | 270 | 270 | 288 | 288 |

Note: (1) The data outside the brackets are coefficients, and the data inside the brackets are t values. (2) Fe_ccd, Fe_qcd demonstrate Driscoll and Kraay standard errors. (3) *, **, and *** indicate the significance of 10%, 5%, and 1%, respectively.

In economically developed regions, the coefficient of PGDP$^3$ in the cubic model is significantly positive, but there is no extreme point. With economic growth, the cultivated land pressure continues to rise. The coefficient of PGDP$^2$ in the square model is significantly positive. With economic growth, the pressure of cultivated land first decreases and then increases. The influence of control variables on cultivated land pressure in developed regions is consistent with basic estimation.

In economically underdeveloped regions, the coefficient of PGDP$^3$ in the cubic model is significantly negative, and there is also no extreme point. As the economy grows, the cultivated land pressure continues to decrease. The estimation result of the squared model shows that the relationship between economic growth and cultivated land pressure in underdeveloped regions is a U-shaped curve. The coefficients of control variables show that their influence direction and significance are different from the regression results with the whole sample. The impact of population growth becomes negative, while the impact of urbanization becomes positive. This might be because the population loss in underdeveloped regions is serious, and the rise in population can increase the agricultural labor force. The effects of effective irrigation, fertilizer application, pesticide input, and agricultural machinery power on cultivated land pressure in underdeveloped regions become insignificant. This shows that the agricultural cultivation technology in underdeveloped regions need to be improved.

Comparing the rebound points of cultivated land pressure in developed regions and underdeveloped regions, it can be found that the rebound point in economically developed regions is larger. This is due to the higher level of agricultural production and technology in developed regions, which delays the rebound of cultivated land pressure.

3.5.2. Different Grain Production and Sales Regions

The estimation results of different grain production and sales regions are shown in Table 11.

From the impact of economic growth on cultivated land pressure, there are differences in distinct grain production and sales regions. The coefficient of PGDP$^3$ in the cubic model is significantly positive in the main sales regions. That is to say, with economic growth, the cultivated land pressure increases firstly, then decreases, and finally increases again. The coefficients of PGDP$^3$ in the cubic model are significantly negative in the production and sales balance regions and the main production regions, and there is no extreme point. As the economy grows, the cultivated land pressure decreases. The coefficients of PGDP$^2$ in the squared model are significantly positive in all regions, and the cultivated land pressure first decreases and then increases with economic growth. The rebound point of cultivated land pressure in the main grain producing regions is much larger than other regions. This shows that the cultivated land in the main production regions has a stronger population support capacity (average cultivated land pressure: production and sales balance regions = 3.686 > main sales regions = 2.514 > main production region = 0.890), which delays the rebound of cultivated land pressure.

The coefficients of the control variables show that the influence direction and significance of a few variables change compared with the basic estimation. The impact of urbanization on cultivated land pressure is positive in the main production regions, but negative in the main grain sales areas. This is because the population urbanization in the main sales regions promotes the improvement of agricultural machinery power and the proportion of grain crops, which eases the cultivated land pressure. However, the high proportion of grain crops planted in the main production regions is highly dependent on labor, and the excessive population loss makes agricultural operations develop in an extensive direction. This is consistent with other scholars' research [76]. The influence of pesticide input on cultivated land pressure is significantly negative in the production and sales balance regions and the main production regions. This is because the pesticide input in these two regions is low (pesticide input per unit of cultivated land: main sales regions = 0.0029 > main production regions = 0.0015 > production and sales balance regions = 0.0005).

**Table 11.** Estimation results of different grain production and sales regions.

| Variables | Main Sales Regions | | Production and Sales Balance Regions | | Main Production Regions | |
|---|---|---|---|---|---|---|
| | FE_ccd | FE_qcd | FE_ccd | FE_qcd | FE_ccd | FE_qcd |
| PGDP$^3$ | 0.076 *** (11.323) | — | −0.223 *** (−3.956) | — | −0.015 * (−1.879) | — |
| PGDP$^2$ | −1.039 *** (−9.227) | 0.299 *** (4.937) | 2.363 *** (4.922) | 0.707 *** (8.364) | 0.222 * (2.150) | 0.042 ** (2.767) |
| PGDP | 4.611 *** (6.919) | −2.760 ** (−2.867) | −9.605 *** (−7.616) | −5.856 *** (−11.789) | −1.311 ** (−2.914) | −0.584 *** (−3.962) |
| POP | 10.370 *** (4.048) | 12.447 *** (5.408) | −0.885 (−0.138) | 7.835 (1.407) | 3.308 *** (4.985) | 4.069 *** (6.525) |
| UR | 0.008 (0.742) | −0.059 * (−1.951) | −0.035 (−1.033) | −0.045 (−1.344) | 0.002 (0.324) | 0.001 (0.272) |
| SI | 0.124 (0.638) | 0.779 ** (3.260) | 0.180 ** (3.018) | 0.189 *** (3.355) | 0.027 ** (2.912) | 0.016 ** (2.466) |
| TI | 0.002 (0.009) | 0.484 * (2.329) | 0.133 ** (2.557) | 0.135 ** (2. 650) | 0.018 (1.480) | 0.010 (0.802) |
| EI | −4.625 *** (−3.664) | −7.054 ** (−3.683) | −4.520 (−1.589) | −4.394 (−1.638) | −0.664 (−1.758) | −1.032 ** (−2.614) |
| FA | −1.031 (−0.045) | 10.732 (0.470) | −3.352 (−0.205) | 6.334 (0.382) | −11.268 *** (−3.134) | −11.395 *** (−3.221) |
| PI | 130.368 (0.564) | −73.659 (−0.371) | −1800.000 *** (−5.121) | −2100.000 *** (−7.533) | −202.606 ** (−2.683) | −193.238 ** (−2.618) |
| MP | −4.592 ** (−2.954) | −3.064 * (−1.968) | 1.040 (1.087) | 0.505 (0.546) | 0.290 (1.733) | 0.261 (1.590) |
| Cons | −5.438 (−0.294) | −39.526 * (−2.061) | 7.640 (0.949) | 2.833 (0.435) | 0.163 (0.194) | −0.227 (−0.295) |
| Time−fixed effect | Yes | Yes | Yes | Yes | Yes | Yes |
| Region−fixed effect | Yes | Yes | Yes | Yes | Yes | Yes |
| R$^2$ | 0.891 | 0.846 | 0.555 | 0.527 | 0.463 | 0.443 |
| F test | 436.79 *** | 85.41 *** | 213.74 *** | 201.87 *** | 779.63 *** | 365.65 *** |
| Curve shape | N | U | Decrement | U | Decrement | U |
| Maximum extreme point | 3.794 | — | — | — | — | — |
| Minimum extreme point | 5.342 | 4.619 | — | 4.144 | — | 6.924 |
| Obs. | 126 | 126 | 198 | 198 | 234 | 234 |

Note: (1) The data outside the brackets are coefficients, and the data inside the brackets are t values. (2) Fe_ccd and Fe_qcd demonstrate Driscoll and Kraay standard errors. (3) *, **, and *** indicate the significance of 10%, 5%, and 1%, respectively.

## 4. Conclusions

Land is of great significance for ensuring food security and promoting economic development. Under the influence of many uncertain factors, such as the COVID-19 pandemic, the Sino–US trade friction, and the Russia–Ukraine conflict, global food security is seriously threatened. The issue of using limited cultivated land resources to guarantee food security and ensure "the rice bowl must be held in our own hands" has become a research hotspot. Based on the cultivated land pressure index and Kuznets curve model, this study analyzes the impact of economic growth on cultivated land pressure. The conclusions are as follows: (1) The relationship between economic growth and cultivated land pressure is an N-shaped or U-shaped curve in China from 2000 to 2017. When the per capita GDP is about 40,000 yuan/person, the cultivated land pressure rebounds. (2) There are regional differences in the impact of economic growth on cultivated land pressure. The per capita GDP at the rebound points of cultivated land pressure in economically developed regions and major grain producing regions are relatively high.

The research of this paper shows that economic growth and cultivated land pressure are sometimes synchronized and sometimes decoupled. With economic growth, the

cultivated land pressure would fluctuate. Cultivated land pressure is affected by many factors, such as population growth, industrial structural changes, technological progress, government policies, and awareness of cultivated land protection. At the current stage, the cultivated land pressure is facing a rebound period from reduction to increase. We should always be vigilant. More attention should be paid to cultivated land protection, and cultivated land pressure should be controlled. Only in this way can we prevent cultivated land pressure from long-term synchronous growth with the economy.

Thus, the following policy recommendations are put forward: (1) We must pay attention to cultivated land protection in the process of economic growth. A decrease in cultivated land pressure is supported by many factors, such as industrial structural changes, technological progress, and increased awareness of cultivated land protection. Only by directing more capital and technology to cultivated land protection in the process of economic development can we effectively control the cultivated land pressure. Some specific measures should be implemented, including improving the compensation system of cultivated land protection, increasing subsidies for the purchase of agricultural machinery, and supporting the development of modern seed industry. (2) We must also prevent an increase of cultivated land pressure caused by urban expansion. By implementing land use control and national land and space planning, the impact of disorderly urban expansion on cultivated land pressure might be weakened. Meanwhile, improving the economical and intensive utilization of urban construction land can reduce the occupation of cultivated land for construction, which might alleviate cultivated land pressure. In practice, it is necessary to strictly delineate and adhere to the control lines of urban development boundaries, permanent basic farmland, and ecological protection. Only in this way can we guide the orderly development of cities and effectively protect cultivated land and ecological environment.

There are some limitations in this study. Firstly, this paper only analyzes the relationship between economic growth and cultivated land pressure at the provincial level, due to the availability of data. However, some provinces have broad jurisdictions, and there are differences in economic growth and cultivated land pressure within the province. Taking cities or counties as the basic research unit can more accurately reflect cultivated land pressure and its influencing factors, which is a research direction worthy of being carried out in the future. Secondly, this paper does not pay attention to the spatial correlation of the cultivated land pressure and its influencing factors. However, grain production and sales, economic development level, and population mobility may have spatial characteristics, which is also a content worthy to study.

**Author Contributions:** Conceptualization, H.Z. and X.W.; methodology, H.Z. and Y.W.; software, Y.W.; validation, H.Z. and X.W.; formal analysis, H.Z. and Y.W.; investigation, H.Z. and Y.W.; resources, H.Z. and X.W.; data curation, Y.W.; writing—original draft preparation, H.Z. and X.W.; writing—review and editing, H.Z., X.W. and Y.W.; visualization, Y.W.; supervision, H.Z. and X.W.; project administration, H.Z. and X.W.; funding acquisition, H.Z. All authors have read and agreed to the published version of the manuscript.

**Funding:** This research was funded by the National Social Science Fund (13BGL149) and the Statistical Development Special Project of Sichuan Social Science Fund (SC19TJ024).

**Institutional Review Board Statement:** Not applicable.

**Informed Consent Statement:** Not applicable.

**Data Availability Statement:** Publicly available datasets were analyzed in this study. These data can be found at: https://data.stats.gov.cn/, https://www.fao.org/faostat/zh/#data/RL, and https://data.cnki.net/Yearbook/Navi?type=type&code=A, accessed on 15 March 2022.

**Conflicts of Interest:** The authors declare no conflict of interest.

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
