# Peer review of "Does Economic Growth Lead to an Increase in Cultivated Land Pressure? Evidence from China"

_land, doi:10.3390/land11091515_

Round 1

Reviewer 1 Report

Dear Authors,

basically, your article is not badly done but it leaves me a bit perplexed, because I think that the objectives of the work should be better explained, some parts should be moved and the part of the results should be better organized. In any case, I very much appreciated the work you have done and here are some of my suggestions, both more general and more specific, I hope they can help you.

In general, there is a problem with the references, you probably made a mistake in inserting them in the program suggested by MDPI because many are not at all up to page 10, but then they are there and they seem correct, so please revise them.

Paragraph 2 I would call Matherials and methods and the sub-paragraph on the study area (which is currently in paragraph 4) I would put it at the beginning of this methodological part, before the actual analysis, because it talks about the regions, how they are divided, their characteristics, etc.

In my opinion, paragraph 4 should be part of the Results, I would start directly with subparagraphs 4.1 and 4.2 and putting the initial part in paragraph 2, as mentioned previously.

Please, try to review the editing in terms of space (even before and after the figures or tables), review the dimensions of the tables trying to align them with the text, in the notes under the tables the line spacing should be reduced, adjust the titles of paragraphs and subparagraphs.

Pease find below some specific suggestions:

·       Line 55 inter-national I believe it should be “international”

·       Line 61-62: “an American economist” I think it can be omitted, also in line 64-65: “an economist at Princeton University” can be omitted

·       Line 71 ex-tended should be “extended”

·       Lines 125-127 please specify when this thing happened, put some date, because otherwise it seems like a contradiction to me

·       Lines 132-133 “With the advancement of agricultural technology” this sentence can be deleted because is a repetition and you can continue directly, simply by putting a comma and not the dot in the previous sentence

·       Line 150-151 “the public” should perhaps be “public entity”?

·       line 161: in the sentence “…may be different in different economic…” try to use a synonym for different, for example you can write “…may be different in distinct economic…”

·       Line 172 seems to me that the font and size are different from the rest, please check it in general

·       Line 184-186 “Cultivated land pressure index was used to characterize the cultivated land pressure. The basic calculation formula of cultivated land pressure index is as follows” I would modify the sentence in this way: “Cultivated land pressure index was used to characterize the cultivated land pressure and the basic calculation formula of it is as follows”

·       Line 266 “there was…” should be “There was…”

·       Line 272 you say that GDP is growing but the unit of measurement is not specified, what are they? Yen? the same in line 308

·       Line 318 I believe that “Bond” is the name of an author

·       Line 350 the title of subsection 4.1 has a too large font, it must be moved towards the center, and I do not think it goes in bold

·       line 336-337: “…. the grain production capacity of different provinces is different” should be : “…. the grain production capacity of different provinces is diversified”

Author Response

Response to Reviewer 1 Comments

Dear reviewers and editor,

Thanks for your important comments on this manuscript. The authors appreciate the constructive comments and technical questions provided by the reviewer. Those comments have been very helpful for the authors to improve the quality of this manuscript. The authors tried to answer the questions in this revision and improve the quality of this paper. The point-to-point responses to the comments by reviewer are listed one by one as follow:

Reviewer 1

Basically, your article is not badly done but it leaves me a bit perplexed, because I think that the objectives of the work should be better explained, some parts should be moved and the part of the results should be better organized. In any case, I very much appreciated the work you have done and here are some of my suggestions, both more general and more specific, I hope they can help you.

Point 1: In general, there is a problem with the references, you probably made a mistake in inserting them in the program suggested by MDPI because many are not at all up to page 10, but then they are there and they seem correct, so please revise them.

Response 1: Thanks for this comment. The reference links in revision have been modified to ensure that they can up to the end of the text.

Point 2: Paragraph 2 I would call Materials and methods and the sub-paragraph on the study area (which is currently in paragraph 4) I would put it at the beginning of this methodological part, before the actual analysis, because it talks about the regions, how they are divided, their characteristics, etc.

In my opinion, paragraph 4 should be part of the Results, I would start directly with subparagraphs 4.1 and 4.2 and putting the initial part in paragraph 2, as mentioned previously.

Response 2: Thanks for this comment. The study area in section 4 of the original manuscript has been moved to section 2. The title of section 2 has been modified to “Materials and methods”, and the subsection 4.1 and 4.2 of the original manuscript has been moved to section 3.

Point 3: Please, try to review the editing in terms of space (even before and after the figures or tables), review the dimensions of the tables trying to align them with the text, in the notes under the tables the line spacing should be reduced, adjust the titles of paragraphs and subparagraphs.

Pease find below some specific suggestions:

Response 3: Thanks for this comment. Problems of editing have been modified in revision.

Point 4: Line 55 inter-national I believe it should be “international”.

Response 4: Thanks for this comment. The word has been modified in revision.

Point 5: Line 61-62: “an American economist” I think it can be omitted, also in line 64-65: “an economist at Princeton University” can be omitted.

Response 5: Thanks for this comment. The sentence has been modified in revision.

Point 6: Line 71 ex-tended should be “extended”.

Response 6: Thanks for this comment. The word has been modified in revision.

Point 7: Lines 125-127 please specify when this thing happened, put some date, because otherwise it seems like a contradiction to me.

Response 7: Thanks for this comment. The time of this thing happened has been explained and the actual data of China has been supplied in revision.

Point 8: Lines 132-133 “With the advancement of agricultural technology” this sentence can be deleted because is a repetition and you can continue directly, simply by putting a comma and not the dot in the previous sentence.

Response 8: Thanks for this comment. The sentence has been modified in revision.

Point 9: Line 150-151 “the public” should perhaps be “public entity”?

Response 9: Thanks for this comment. The sentence has been modified in revision.

Point 10: line 161: in the sentence “…may be different in different economic…” try to use a synonym for different, for example you can write “…may be different in distinct economic…”.

Response 10: Thanks for this comment. Synonyms of words have been tried to use in revision.

Point 11: Line 172 seems to me that the font and size are different from the rest, please check it in general.

Response 11: Thanks for this comment. The reason why the font and size here are different from others is that the formula is edited with MathType software. The formats of all formulas in this paper have been modified to make them look the same as the rest in revision.

Point 12: Line 184-186 “Cultivated land pressure index was used to characterize the cultivated land pressure. The basic calculation formula of cultivated land pressure index is as follows” I would modify the sentence in this way: “Cultivated land pressure index was used to characterize the cultivated land pressure and the basic calculation formula of it is as follows”.

Response 12: Thanks for this comment. The sentence has been modified in revision.

Point 13: Line 266 “there was…” should be “There was…”.

Response 13: Thanks for this comment. The word has been modified in revision.

Point 14: Line 272 you say that GDP is growing but the unit of measurement is not specified, what are they? Yen? the same in line 308.

Response 14: Thanks for this comment. The unit of measurement has been specified as RMB (yuan).

Point 15: Line 318 I believe that “Bond” is the name of an author 298

Response 15: Thanks for this comment. The word has been modified in revision.

Point 16: Line 350 the title of subsection 4.1 has a too large font, it must be moved towards the center, and I do not think it goes in bold.

Response 16: Thanks for this comment. The title formats of subsection 4.1 and 4.2 have been modified and they have been moved to section 3.

Point 17: line 336-337: “…. the grain production capacity of different provinces is different” should be: “…. the grain production capacity of different provinces is diversified”.

Response 17: Thanks for this comment. The sentence has been modified in revision.

Reviewer 2 Report

This is an interesting study on the relationship between the economic growth and the pressure on cultivated lands. However, certain improvements could be introduced to increase the contribution of the paper to the literature and the land-use agenda.

I would disagree that economic growth is necessarily associated with the increase in the population. Probably, the author refers to the case of China, but even China has not experienced such a growth for a number of recent years. This statement should be reconsidered (see, for example, the abstract and the introduction). the more promising venue is the link between economic growth and the increase in the living standards, i.e., the intensification of the agricultural production and the higher pressure on agricultural lands due to urbanization and industrialization. The author should clearly state what is economic growth (for the purpose of this paper). It is GDP growth, or the improvement in the living standards, or the growth in the industrial production, or anything else. This is critically important for establishing the proper linkages between the variables, as well as for interpreting the results.

Line 15: please specify the country. Also, the author should briefly explain the specific relevance of the land-growth issue for China. 

Line 39: please provide the USD estimates for yuan, it would be helpful for international comparisons.

Line 88: please discuss the current threats to the international food security, refer to the situation in countries other than China, and compare the situation. What is the role of land use in China in this situation and, reversely. how does the international situation affects the land use in China?

Line 101: regional differences issue should be discussed in the introduction. It first appears here in the problem statement section. It is not clear why this issues matters for China and for other countries. International comparisons should be provided.

Line 153: it is not clear what the author means here - what is staged? How does it proceed from the pure theoretical review above? Please bring the theoretical discussion closer to the practical land-use issue addressed in the paper. This is the pure theory that does not contribute much to the current study.

Section 2.2.1: the author should critically discuss the selection of the model. Why and how does this model benefits the particular study? What are the alternatives? Has it been employed previously? What were the findings and limitations? The critical approach is needed to demonstrate the benefits and advantages of the model for this particular study. 

Table 1: all these variables should be discussed in relation to China. It is not clear why these particular variables are selected. The author should explain the importance of each parameter for China.

The quality of the language and grammar must be improved substantially.

Author Response

Response to Reviewer 2 Comments

Dear reviewers and editor,

Thanks for your important comments on this manuscript. The authors appreciate the constructive comments and technical questions provided by the reviewer. Those comments have been very helpful for the authors to improve the quality of this manuscript. The authors tried to answer the questions in this revision and improve the quality of this paper. The point-to-point responses to the comments by reviewer are listed one by one as follow:

Reviewer 2

This is an interesting study on the relationship between the economic growth and the pressure on cultivated lands. However, certain improvements could be introduced to increase the contribution of the paper to the literature and the land-use agenda.

Point 1: I would disagree that economic growth is necessarily associated with the increase in the population. Probably, the author refers to the case of China, but even China has not experienced such a growth for a number of recent years. This statement should be reconsidered (see, for example, the abstract and the introduction). the more promising venue is the link between economic growth and the increase in the living standards, i.e., the intensification of the agricultural production and the higher pressure on agricultural lands due to urbanization and industrialization. The author should clearly state what is economic growth (for the purpose of this paper). It is GDP growth, or the improvement in the living standards, or the growth in the industrial production, or anything else. This is critically important for establishing the proper linkages between the variables, as well as for interpreting the results.

Response 1: Thanks for this comment. We accept the reviewer’s opinion that population growth is not necessarily associated to economic growth. The original manuscript incorrectly stated the relationship between economic growth and population growth, because authors observed that China's economy and population grew rapidly during the period from the reform and opening up to the end of the 20th century (the annual average growth rate of GDP from 1978 to 2000 was 9.78%, and the annual average natural growth rate of population was 12.47 ‰). However, in recent years, China's economy has maintained rapid growth, while the natural growth rate of population has declined rapidly (in 2021, the growth rate of GDP was 8.1%, while the natural growth rate of population was 0.34 ‰). Thank you again for your comment. We have modified the statement in revision. The authors agree the opinion that economic growth, mainly GDP and income growth, improves living standards and changed the food consumption structure, thus the food demand increases, which might affect cultivated land pressure. In this paper, economic growth refers to GDP growth, due to the purpose of this paper is to analyze the impact of macroeconomic growth on the cultivated land pressure. In the empirical analysis, per capita GDP growth is used to express economic growth.

Data source: website of China National Bureau of Statistics https://data.stats.gov.cn/easyquery.htm?cn=C01

Fukase, E., Martin, W. Economic Growth, Convergence, and World Food Demand and Supply. World Development 2020, 132, 104954. https://doi.org/10.1016/j.worlddev.2020.104954

Gandhi, V.P., Zhou, Z.Y. Food Demand and the Food Security Challenge with Rapid Economic Growth in the Emerging Economies of India and China. Food Research International 2014, 63, 108-24. https://doi.org/10.1016/j.foodres.2014.03.015

Dang, G., Pheng, L.S. Theories of Economic Development. Infrastructure Investments in Developing Economies 2014,11–26. https://doi.org/10.1007/978-981-287-248-7_2

Point 2: Line 15: please specify the country. Also, the author should briefly explain the specific relevance of the land-growth issue for China.

Response 2: Thanks for this comment. the country has been specified as China, and specific relevance of the land-growth issue for China is explained in revision. Due to the economic growth and urban expansion in China since 1978 led to the loss of a large amount of cultivated land, the contradiction between “economic growth” and “food security” has become increasingly prominent.

Point 3: Line 39: please provide the USD estimates for yuan, it would be helpful for international comparisons.

Response 3: Thanks for this comment. The USD estimates for yuan has been provided in revision.

Point 4: Line 88: please discuss the current threats to the international food security, refer to the situation in countries other than China, and compare the situation. What is the role of land use in China in this situation and, reversely. how does the international situation affect the land use in China?

Response 4: Thanks for this comment. The analysis of threats to international food security has been added to paragraph 3 in the introduction. Under such international situation, China needs to use its own land to feed its own population, and it is increasingly important to protect cultivated land and ensure its population support capacity.

Point 5: Line 101: regional differences issue should be discussed in the introduction. It first appears here in the problem statement section. It is not clear why this issue matters for China and for other countries. International comparisons should be provided.

Response 5: Thanks for this comment. The grain supply capacity in different regions of China is diverse. Regions with developed economic and high grain production have stronger grain supply capacity and greater grain supply flexibility, and the pressure of cultivated land population support may be less affected by economic growth. In the world, under the unstable international situation, trade is restricted, and nations relying on imports are vulnerable to food supply shocks. The discussion of regional differences would help to put forward policy suggestions for coordinating economic growth and food security pertinently.

Udmale, P., Pal, I., Szabo, S., Pramanik, M., Large, A. Global Food Security in the Context of COVID-19: A Scenario-based Exploratory Analysis. Progress in Disaster Science 2020, 7, 100120. https://doi.org/10.1016/j.pdisas.2020.100120

Point 6: Line 153: it is not clear what the author means here - what is staged? How does it proceed from the pure theoretical review above? Please bring the theoretical discussion closer to the practical land-use issue addressed in the paper. This is the pure theory that does not contribute much to the current study.

Response 6: Thanks for this comment. “Staged” in the original manuscript means that the impact of economic growth on cultivated land pressure is sometimes positive and sometimes negative, and the impact is diverse at different stages of economic growth. The meaning may be unclear due to language and grammar problems. The sentence has been modified in revision. In order to increase the practicability of the theoretical analysis, the data reflecting the impact of economic growth on China's cultivated land use has been appropriately added in the theoretical analysis.

Point 7: Section 2.2.1: the author should critically discuss the selection of the model. Why and how does this model benefits the particular study? What are the alternatives? Has it been employed previously? What were the findings and limitations? The critical approach is needed to demonstrate the benefits and advantages of the model for this particular study.

Response 7: Thanks for this comment. Some contents have been added in “2.3.1 Model setting” to explain these problems. Firstly, the correlation between Environmental Kuznets curve (EKC) model and this research has been described. Secondly, a brief introduction to Environmental Kuznets curve (EKC) model and its application has been added, and the advantages and limitations of the model have been analyzed. Finally, the benefits and advantages of EKC model for our study and how to deal with the limitations of the model have been introduced.

Point 8: Table 1: all these variables should be discussed in relation to China. It is not clear why these particular variables are selected. The author should explain the importance of each parameter for China.

Response 8: Thanks for this comment. The reasons for the selection of variables and their importance for China have been explained in revision.

Point 9: The quality of the language and grammar must be improved substantially.

Response 9: Thanks for this comment. The authors have carefully checked the whole paper and the quality of the language and grammar has been improved.

Reviewer 3 Report

About the submission with the title "Does economic growth lead to cultivated land pressure increases? Evidence from China" I have the folowing explanations:

The abstract should highlight better the objectives and methodologies considered.

The paper has several errors in citations and references, what hampers the understanding of the several sources.

The interrelationships of the models considered with the Kuznets curve need deeper explanations. Maybe, it could be important first explain briefly in this section the Kuznets curve and after how the models considered were obtained.

The results presented for the several estimations should be assessed for potential problems related with multicollinearity, heteroscdasticity, autocorrelation (including spatial), endogeneity, linearity, normality, adequacy of the model, ...

In the conclusions section, the practical implications, policy recommendations and directions for future research should be clear.

Author Response

Response to Reviewer 3 Comments

Dear reviewers and editor,

Thanks for your important comments on this manuscript. The authors appreciate the constructive comments and technical questions provided by the reviewer. Those comments have been very helpful for the authors to improve the quality of this manuscript. The authors tried to answer the questions in this revision and improve the quality of this paper. The point-to-point responses to the comments by reviewer are listed one by one as follow:

Reviewer 3

About the submission with the title "Does economic growth lead to cultivated land pressure increases? Evidence from China" I have the following explanations:

Abstract

Point 1: The abstract should highlight better the objectives and methodologies considered.

Response 1: Thanks for this comment. More information about the objectives and methodologies has been added into the abstract.

Introduction

Point 2: The paper has several errors in citations and references, what hampers the understanding of the several sources.

Response 2: Thanks for this comment. The reference links in revision have been modified to ensure that they can up to the end of the text.

Models

Point 3: The interrelationships of the models considered with the Kuznets curve need deeper explanations. Maybe, it could be important first explain briefly in this section the Kuznets curve and after how the models considered were obtained.

Response 3: Thanks for this comment. The interrelationships between the model and Kuznets curve have been explained detailly in “2.3.1. Model settings”. And the introduction of Kuznets curve and the reason for building the model are added in revision.

Results

Point 4: The results presented for the several estimations should be assessed for potential problems related with multicollinearity, heteroscedasticity, autocorrelation (including spatial), endogeneity, linearity, normality, adequacy of the model, ...

Response 4: Thanks for this comment.

Normality and multicollinearity tests of the data have been provided in “2.4. Data sources”. The skewness and kurtosis are added in Table 3. The variance expansion factor (VIF) is provided in Table 4. The VIF values of all variables are less than 10, thus there is no multicollinearity.

As for the heteroscedasticity, correlation and adequacy of the model, the Modified Wald test, Frees test, Wooldridge test, F test, F statistic, Hausman test have been carried out, and the results are shown in the table of regression results. The results show that the model has heteroscedasticity, cross-sectional correlation and autocorrelation. Therefore, the feasibility generalized least squares (FGLS) and Driscoll and Kraay standard error are employed. F test, F statistic, and Hausman test proved the adequacy of model, which indicate that the fixed-effects model is better than the mixed OLS and random-effects model.

As for the spatial correlation of the model, this paper did not consider the spatial relationship, so it is not tested. But the reviewer provides an aspect worth studying. In the future, we will research the spatial relationship between cultivated land pressure and its influencing factors.

The endogeneity of the model is avoided by using the generalized moment estimation (GMM). The influence of explanatory variables and control variables after considering the endogeneity is basically consistent with the benchmark regression.

Conclusions

Point 5: In the conclusions section, the practical implications, policy recommendations and directions for future research should be clear.

Response 5: Thanks for this comment. More operational policy recommendations have been put forward in the conclusions section. The limitations of this paper and the directions for future research have been added in revision.

Author Response

Response to Reviewer 4 Comments

Dear reviewers and editor,

Thanks for your important comments on this manuscript. The authors appreciate the constructive comments and technical questions provided by the reviewer. Those comments have been very helpful for the authors to improve the quality of this manuscript. The authors tried to answer the questions in this revision and improve the quality of this paper. The point-to-point responses to the comments by reviewer are listed one by one as follow:

Reviewer 4

This paper uses the cultivated land pressure index to represent the cultivated land population support pressure, and explores the relationship between economic growth and cultivated land pressure based on the panel data of 31 provinces from 2000 to 2017. The author found that he impacts of economic growth on cultivated land pressure are N-shaped or U-shaped curves. It is an interesting result.

However, I have some reservations regarding the materials and methods. I will show my main concerns as follows, and I guess the paper should be considered only if the author can well solve these problems.

Point 1: For the Theoretical analysis, the author mentioned that “Economic structural changes, technological progress, government policy improvement, and changes in residents’ preferences brought about by economic growth can all affect the cultivated land pressure. “Why does the author explain the causes of cultivated land pressure from these four aspects?

Response 1: Thanks for this comment. Combining the causes of the environmental Kuznets curve and the factors that affect the loss of cultivated land, this paper explains the impact of economic growth on cultivated land pressure from four aspects. A brief description for the selection of the influence analysis angle has been provided in revision.

Kijima, M., Nishide, K., Ohyama, A. Economic Models for the Environmental Kuznets Curve: A Survey. Journal of Economic Dynamics & Control 2010, 34(7), 1187-201. https://doi.org/10.1016/j.jedc.2010.03.010

Qu, F.T., Wu, L.M. Hypothesis and Validation on the Kuznets Curves of Economic Growth and Farmland Conversion. Resources Science 2004, 26, 61-67. https://doi.org/10.3321/j.issn:1007-7588.2004.05.009

Point 2: For the variable’s selection, it is suggested that the author should explain the reason or basic of such a measure for cultivated land pressure index. Moreover, what is the minimum per capita cultivated land area? How to define it? Likewise, the author needs to explain the meaning of the letter in the formula in detail.

Response 2: Thanks for this comment. The reason for adopting the cultivated land pressure index has been added in “2.3.2. Variables selection”. The minimum per capita cultivated land area refers to the area of cultivated land required to ensure the normal food consumption of each person under a certain level of grain self-sufficiency and cultivated land production capacity in a certain region. The meaning of the letters in the formula has been explained detailly in revision.

Point 3: Other problems: Many language errors are in the context. Please check them carefully. For example, reference source not found

Response 3: Thanks for this comment. The authors have carefully checked the whole paper and the quality of the language and grammar has been improved. The reference links in revision have been modified to ensure that they can up to the end of the text.

Reviewer 5 Report

The paper is interesting, but it needs to be improved further. The errors in the citations caused me several doubts.

Main remarks:

Introduction-[Error! Reference source not found.]-Instead of the references this error ocrrued. I'm not able to identify the respective references to do my analysis. This also ocurred in the rest of the paper.

When correcting this please consider verifying if all rellevant references at international level are presented.

L90-"Existing studies"-When correcting the text, you must mention them.

The theoretical approach seems to be constente, bu I have several doubts because the lack of proprer references

L199-"Referring to Zhu (2016)"-When correcting the references, please aloso correct this one.

"Table 2. Descriptive statistics of variables"-OK well presented. Besides this a correlation matrix of the variabales could also be presented.

 L279-"Urbanization rate had no significant effect on cultivated land pressure"-If possible a more complete discussion regarding this issue could be added.

L335-"regions with different economic development"-Divided using what data form what sources?

Author Response

Response to Reviewer 5 Comments

Dear reviewers and editor,

Thanks for your important comments on this manuscript. The authors appreciate the constructive comments and technical questions provided by the reviewer. Those comments have been very helpful for the authors to improve the quality of this manuscript. The authors tried to answer the questions in this revision and improve the quality of this paper. The point-to-point responses to the comments by reviewer are listed one by one as follow:

The paper is interesting, but it needs to be improved further. The errors in the citations caused me several doubts.

Main remarks:

Point 1: -[Error! Reference source not found.]- Instead of the references this error occurred. I'm not able to identify the respective references to do my analysis. This also occurred in the rest of the paper.

Response 1: Thanks for this comment. The reference links in revision have been modified to ensure that they can up to the end of the text.

Point 2: When correcting this please consider verifying if all relevant references at international level are presented.

Response 2: Thanks for this comment. The relevant references at international level have been presented in revision.

Point 3: L90-"Existing studies"-When correcting the text, you must mention them.

Response 3: Thanks for this comment. The references of "Existing studies" have been mentioned in revision.

Point 4: The theoretical approach seems to be contented, but I have several doubts because the lack of proper references.

L199-"Referring to Zhu (2016)"-When correcting the references, please also correct this one.

Response 4: Thanks for this comment. The reference link of “Zhu (2016)” has been corrected in revision.

Point 5: "Table 2. Descriptive statistics of variables"-OK well presented. Besides this a correlation matrix of the variables could also be presented.

Response 5: Thanks for this comment. A correlation matrix of the variables has been presented in revision.

Point 6: L279-"Urbanization rate had no significant effect on cultivated land pressure" -If possible, a more complete discussion regarding this issue could be added.

Response 6: Thanks for this comment. A more complete discussion regarding the impact of urbanization on cultivated land pressure has been added in revision.

Point 7: L335-"regions with different economic development"-Divided using what data form what sources?

Response 7: Thanks for this comment. The provinces are divided into developed regions and undeveloped regions based on the median of the average per capita GDP from 2000 to 2017. The data is obtained from the “China Statistical Yearbook (2001–2018)”. The basis for regional division is explained in revision.

Tang, P., Feng, Y., Li, M., Zhang, Y. Y. Can the performance evaluation change from central government suppress illegal land use in local governments? A new interpretation of Chinese decentralisation. Land Use Policy 2021, 108,105578. https://doi.org/10.1016/j.landusepol.2021.105578

Round 2

Reviewer 2 Report

My Round 1 recommendations have been addressed adequately

Author Response

Response to Reviewer 2 Comments

Reviewer 2

My Round 1 recommendations have been addressed adequately

Dear reviewer,

Thanks a lot for your important comments on the revision, it is very effective in improving the quality of the paper. The authors have improved the quality of this paper again. The language and grammatical problems in this manuscript have been proofread. The background, research design, methods, results, conclusions and references of this manuscript have been sorted and improved.

Reviewer 3 Report

Authors improved the paper.

Author Response

Response to Reviewer 3 Comments

Reviewer 3

Authors improved the paper.

Dear reviewer,

Thanks a lot for your important comments on the revision, it is very effective in improving the quality of the paper. The authors have improved the quality of this paper again. The language and grammatical problems in this manuscript have been proofread. The background, research design, methods, results, conclusions and references of this manuscript have been sorted and improved.

Reviewer 5 Report

The authors corrected the paper according to my recommendations. Good work!

Author Response

Response to Reviewer 5 Comments

Reviewer 5

The authors corrected the paper according to my recommendations. Good work!

Dear reviewer,

Thanks a lot for your important comments on the revision, it is very effective in improving the quality of the paper. The authors have improved the quality of this paper again. The language and grammatical problems in this manuscript have been proofread. The background, research design, methods, results, conclusions and references of this manuscript have been sorted and improved.
